# How Does *Lagenaria siceraria* (Bottle Gourd) Metabolome Compare to *Cucumis sativus* (Cucumber) F. Cucurbitaceae? A Multiplex Approach of HR-UPLC/MS/MS and GC/MS Using Molecular Networking and Chemometrics

**DOI:** 10.3390/foods12040771

**Published:** 2023-02-10

**Authors:** Radwa H. El-Akad, Mohamed G. Sharaf El-Din, Mohamed A. Farag

**Affiliations:** 1Pharmacognosy Department, Pharmaceutical and Drug Industries Institute, National Research Centre, Cairo 12622, Egypt; 2Pharmacognosy Department, Faculty of Pharmacy, Port Said University, Port Said 42515, Egypt; 3Pharmacognosy Department, Faculty of Pharmacy, Cairo University, Cairo 11562, Egypt

**Keywords:** aroma, bottle gourd, chemometrics, cucumber, *Cucumis sativus*, *Lagenaria siceraria*, metabolomics, nutrients

## Abstract

Cucurbitaceae comprises 800 species, the majority of which are known for their nutritive, economic, and health-promoting effects. This study aims at the metabolome profiling of cucumber (*Cucumis sativus*) and bottle gourd (*Lagenaria siceraria*) fruits in a comparative manner for the first time, considering that both species are reported to exhibit several in-common phytochemical classes and bioactivities. Nevertheless, bottle gourd is far less known and/or consumed than cucumber, which is famous worldwide. A multiplex approach, including HR-UPLC/MS/MS, GNPS networking, SPME, and GC/MS, was employed to profile primary and secondary metabolites in both species that could mediate for new health and nutritive aspects, in addition to their aroma profiling, which affects the consumers’ preferences. Spectroscopic datasets were analyzed using multivariate data analyses (PCA and OPLS) for assigning biomarkers that distinguish each fruit. Herein, 107 metabolites were annotated in cucumber and bottle gourd fruits via HR-UPLC/MS/MS analysis in both modes, aided by GNPS networking. Metabolites belong to amino acids, organic acids, cinnamates, alkaloids, flavonoids, pterocarpans, alkyl glycosides, sesquiterpenes, saponins, lignans, fatty acids/amides, and lysophospholipids, including several first-time reported metabolites and classes in Cucurbitaceae. Aroma profiling detected 93 volatiles presented at comparable levels in both species, from which it can be inferred that bottle gourds possess a consumer-pleasant aroma, although data analyses detected further enrichment of bottle gourd with ketones and esters versus aldehydes in cucumber. GC/MS analysis of silylated compounds detected 49 peaks in both species, including alcohols, amino acids, fatty acids/esters, nitrogenous compounds, organic acids, phenolic acids, steroids, and sugars, from which data analyses recognized that the bottle gourd was further enriched with fatty acids in contrast to higher sugar levels in cucumber. This study provides new possible attributes for both species in nutrition and health-care fields based on the newly detected metabolites, and further highlights the potential of the less famous fruit “bottle gourd”, recommending its propagation.

## 1. Introduction

Family Cucurbitaceae comprises 130 genera and 800 species, including several crucial crops of high nutritive and medicinal values and increasing economic interest [1,2]. Roots and seeds of most Cucurbitaceae members share the presence of toxic triterpenes, cucurbitacins. In contrast, their fruits are safe, edible, nutritious, and possess health-promoting effects [1,3]. *Cucumis sativus* (cucumber) is a well-known species of Cucurbitaceae that is cultivated worldwide, and its fruit is edible either fresh or cooked [1,4]; in contrast, the less famous *Lagenaria siceraria* species (bottle gourd) is commonly used on the Indo-Pakistan subcontinent and few other countries in Europe and Africa for its nutritive and therapeutic values [5]. Aside from the macromorphological resemblance, both species are reported to exhibit anti-inflammatory, antioxidant, antimicrobial, anticancer, antihyperlipidemic, and cardioprotective activities, and are used in treatment of GIT disorders [1,4]; additionally, fruits’ extracts are involved in the treatment of skin disorders [3,4,6]. Previous phytochemical screening of *L. siceraria* and *C. sativus* indicated the presence of flavonoids, sterols, terpenes, phenolic acids, saponins, and carbohydrates in both fruits, whereas alkaloids were detected in *C. sativus* [3,4,7].

Despite that both bottle gourd and cucumber fruits share several common nutritional and medicinal purposes as well as their involvement in Ayurvedic and folk medicines [4,8], few studies have reported on their metabolome heterogeneity and how the bottle gourd compares to the most famous fruit in F. Cucurbitaceae, the cucumber, as analyzed using metabolomics and chemometrics. In this study, the first comparative metabolome profiling of *L. siceraria* and *C. sativus* fruit crude extracts is presented as measured via high resolution-ultrahigh performance liquid chromatography coupled to mass spectrometry (HR-UPLC/MS/MS) in both ionization modes for a comprehensive overview of metabolites. HR-UPLC/MS/MS is the analytical tool of choice for rapid and sensitive monitoring of secondary metabolites due to its robustness and selectivity [9]. Identification of metabolites was aided by Global Natural Products Social molecular networking (GNPS), that allowed rapid dereplication of compounds and extrapolating tentative identification into the unknown [9,10].

Solid phase microextraction (SPME) of volatile constituents alongside silylation of nonvolatile compounds followed by GC/MS analyses provided insight of their aroma and nutrients profile, respectively, which affect food value and consumer preferences, likewise for the first time [11]. Owing to the complexity of the obtained datasets via GC/MS analyses, multivariate data analyses using principal component analysis (PCA) and orthogonal projection to latent structures-discriminant supervised data analysis (OPLS) were applied for the first time for samples classification and assigning discriminating metabolites for each fruit.

The study provides new insights into the chemical composition of cucumber and the less investigated species, bottle gourd, in a comparative approach leading to the identification of several first-time-reported metabolites and classes in both species and a better rationalization of their health effects and potential application as nutraceuticals in the future, based on such chemical profiling.

## 2. Materials and Methods

### 2.1. Plant Material

The fresh fruits of cucumber (*Cucumis sativus*) and bottle gourd (*Lagenaria siceraria var. siceraria*) were obtained from Barrage Experimental Farm of Horticulture Research Station, El-Kanater El-Khyreia, Qalubia Governorate, Egypt. They were identified by Prof. Dr. Mahmoud Kotb Hatem; Breeding Research Department for Vegetable Crops, Medicinal, and Aromatic Plants, Horticulture Research Institute, Agricultural Research Center.

### 2.2. Chemicals and Fibers

SPME fiber of stableflex coated with divinylbenzene/carboxen/polydimethylsiloxane (DVB/CAR/PDMS, 50/30 µm) was purchased by Supelco (Oakville, ON, Canada). All chemicals and standards were purchased from Sigma Aldrich (St. Louis, MO, USA). Acetonitrile and formic acid (LC–MS grade) were obtained from J. T. Baker (The Netherlands), and milliQ water was used for UPLC/PDA/ESI–qTOF-MS analysis.

### 2.3. Extracts Preparation for UPLC-MS Analysis

Freeze-dried samples of *C. sativus* and *L. siceraria* fruits (n = 3 each) were ground in liquid nitrogen (Sigma-Aldrich, St. Louis, MO, USA, purity ≥ 99.998%) using pestle and mortar. The extraction procedure was conducted as previously described in Hegazi, N.M. et al., [9]. About 150 of the powdered samples was mixed with 6 mL methanol containing 1 g/mL umbelliferone as an internal standard (Sigma-Aldrich, St. Louis, MO, USA, purity ≥ 98.0%) and homogenized with an Ultra-Turrax (IKA, Staufen, Germany) at 11,000 rpm, 5 × 60 s with 1 min break intervals. Extracts were then vortexed for 1 min, centrifuged at 3000× *g* for 30 min, and filtered through a 22 m pore size filter.

### 2.4. UPLC-MS Analysis

Chromatographic separation was performed on an ACQUITY UPLC system (Waters, Milford, MA, USA) equipped with a HSS T3 column (100 × 1.0 mm, particle size 1.8 µm; Waters). The analysis was carried out under exact conditions described in Hegazi, N.M. et.al., [9]. Characterization of compounds was performed by generation of the elemental formula (mass accuracy < 5 ppm) and considering RT, MS2 data and reference literature.

### 2.5. Molecular Networking

The molecular network (MN) was constructed using the UPLC-HRMS/MS data (in the negative and positive ion mode) from both fruit extracts as prepared in Section 2.3 following exact conditions mentioned in Hegazi, N.M. et al. [9]. The MN parameters were as follows: minimum cosine score 0.70; 0.1 Da parent mass tolerance, 0.5 Da as fragment ion tolerance to create consensus spectra, more than 6 matched peaks, and a minimum cluster size of 2. All of the matches kept between network spectra and library spectra were required to have a score above 0.7 and at least 4 matched peaks. Cytoscape (ver. 3.8.2.) was used for network visualization and analysis.

### 2.6. Headspace Volatiles Analysis of C. sativus and L. siceraria

The sample was prepared and analyzed following the same procedure and conditions reported in Farag et al. [12]. GC–MS analysis was adopted on an Agilent 5977B GC/MSD (Santa Clara, CA, USA) equipped with a DB-5 column (30 m × 0.25 mm i.d. × 0.25 µm film thickness; Supelco, Bellefonte, PA, USA) and coupled to a quadrupole mass spectrometer following the exact conditions mentioned in Farag, M.A. et al., [12]. For assessment of replicates, three different samples for each fruit were analyzed under the same conditions. Blank runs were conducted during sample analyses. The mass spectrometer was adjusted to EI mode at 70 eV with a scan range set at *m*/*z* 40–500.

### 2.7. GC–MS Analysis of Silylated Primary Metabolites in C. sativus and L. siceraria Fruits

100 Mg of finely freeze-dried powdered sample (for both fruits) was extracted with 5 mL 100% methanol with sonication for 30 min using Branson CPX-952-518R set at 36 °C, (Branson Ultrasonics, Carouge, SA Switzerland.) and with regular shaking, followed by centrifugation (LC-04C 80-2C regen lab prp centrifuge, Zhejiang, China) at 12,000× *g* for 10 min to eliminate debris. For evaluation of biological replicates, 3 independent samples for each fruit was analyzed under the same conditions. Then, 100 µL of the methanol extract was kept in opened screw-cap vials and left to evaporate under stream of nitrogen gas until full dryness. For derivatization, 150 µL of N-methyl-N-(trimethylsilyl)-trifluoroacetamide (MSTFA), previously diluted 1/1 with anhydrous pyridine, was mixed with the dried methanol extract and incubated (Yamato Scientific DGS400 Oven, QTE TECHNOLOGIES, Hanoi, Vietnam) for 45 min at 60 °C previous analysis using GC–MS. Separation of silylated derivatives was completed on a Rtx-5MS Restek, Bellefonte, PA, USA (30-m length, 0.25-mm inner diameter and 0.25-m film). Analysis of these primary metabolites followed the exact protocol detailed in Sedeek, M.S. et al., and Farag, M.A. et al. [11,12].

### 2.8. Metabolites Identification and Multivariate Data Analyses of Volatile and Non-Volatile Silylated Components Analyzed Using GC/MS

Identification was performed by comparing their retention indices (RI) in relation to n-alkanes (C8-C30), mass matching to NIST, WILEY library database and with standards if available. Peaks were first deconvoluted using AMDIS software (www.amdis.net, accessed on 23 April 2022) before mass spectral matching. Peak abundance data were exported for multivariate data analysis by extraction using MET-IDEA software (Broeckling, Reddy, Duran, Zhao, and Sumner, 2006). Data were then normalized to the amount of spiked internal standard (Z)-3-hexenyl acetate, pareto scaled and then subjected to principal component analysis (PCA), hierarchical clustering analysis (HCA) and partial least squares discriminant analysis (OPLS-DA) using SIMCA-P version 13.0 software package (Umetrics, Umeå, Sweden). All variables were mean-centered and scaled to Pareto variance.

## 3. Results and Discussion

### 3.1. Metabolome Profiling of L. siceraria and C. sativus Fruit Extracts via HR-UPLC/PDA/ESI-MS Based Molecular Networking

The comparative profiling of metabolites in bottle gourd and cucumber crude fruit methanol extracts was conducted via HR-UPLC/MS/MS in both negative and positive ionization modes (Section 2.4) for a comprehensive overview of metabolites that belong to different phytochemical classes. Gradient elution system of formic acid in water (0.1%): acetonitrile allowed for metabolites’ elution in order of their decreasing polarity. The obtained base peak chromatograms (BPCs) of each fruit extract in both ionization modes are presented in Figure 1.

Furthermore, molecular networking was applied herein for the in-depth exploration and discrimination of samples guided by the Global Natural Products Social (GNPS) networking software and its spectral library that aided in peaks’ assignment and annotation based on analyzing the HR-tandem MS/MS data [9,10]. Two molecular networks (MN) were constructed from both ionization mode analyses that provided visual discrimination of samples based on metabolites’ abundance in each node, presented as a pie chart, allowing for rapid dereplication of known compounds (Figure 2 and Appendix A). Samples were coded with orange and green colors for bottle gourd and cucumber, respectively. All nodes were labelled with parent mass and edges were labelled with neutral loss values.

Identification was based on determining retention time (Rt. min.) for each metabolite, its mass spectral data—including molecular ion, daughter ions, their respective formulae and fragmentation pattern—and comparing the collective data with the reported literature and databases such as HMDB, PubChem, FooDB, the Phytochemical dictionary of natural products, and others, combined with GNPS library annotations. The clustering of metabolites in MN (minimum two connected nodes) was based on shared fragments and their fragmentation pattern; this allowed us to extrapolate the identification of annotated compounds to the unknown peaks aided by the generated formulae [10].

Overall, 107 peaks were annotated in both modes (Table 1), belonging to amino acids, organic acids, cinnamates, alkaloids, flavonoids, pterocarpans, alkyl glycosides, sesquiterpenes, saponins, lignans, fatty acids/amides, and lysophospholipids, including several first-time reported metabolites and classes, as explained in detail in the following subsections. The constructed MN from the HR- negative ESI-MS/MS analysis was composed of 351 nodes and 449 edges, in which the clusters of interest included cluster A: fatty acids/amides, cluster B: lysophospholipids, cluster C: flavonoids and pterocarpans, cluster D: alkyl glycosides, cluster E: saponins, cluster F: organic acids, cluster G: amino acids derivatives, cluster H: sesquiterpenes, cluster I: lysophosphatidic acid derivatives, and cluster J: cinnamates (Figure 2). The positive MN was composed of 1002 nodes and 1451 edges, in which the clusters of interest included cluster A: lignans, cluster B: flavonoids, cluster C: amino acid derivatives, singleton D: alkaloid, and cluster E: fatty acids/amides (Appendix A).

The detection of alkaloids in positive mode versus phenolic acids in negative mode is expected considering the improved sensitivity for each class in respective mode and highlighting the importance of acquiring in different ionization types. Glycosides were detected based on the neutral loss of the attached *O*-sugar moieties at 162 amu (C_6_H_10_O_5_) for hexoses, 146 amu (C_6_H_10_O_4_) for rhamnose, 176 amu (C_6_H_8_O_6_) for glucuronide and 132 amu (C_5_H_8_O_4_) for pentoses. In some spectra, peaks detected at *m*/*z* 113 (C_5_H_5_O_3_^−^), *m*/*z* 101 (C_4_H_5_O_3_^−^), *m*/*z* 71 (C_3_H_3_O_2_^−^) and/or *m*/*z* 59 (C_2_H_3_O_2_^−^) correspond to hexose fragments [13]. Acylation of glycosides with acetyl or malonyl moieties was recognized by additional mass and/or neutral loss of 42 amu (C_2_H_2_O) or 86 amu (C_3_H_2_O_3_), respectively, while *C*-glycosides showed significant neutral losses of 90 amu and 120 amu resulting from 0,2 and 0,3-sugar ring cleavage [14].

#### 3.1.1. Amino Acids and Amines

Several amino acids and amine derivatives were eluted early, as detected in chromatograms at Rt. 4–11 min. (Figure 1), and grouped in two major clusters, i.e., G and C, in negative and positive MNs, respectively (Figure 2 and Appendix A), from which 14 metabolites could be annotated in peaks 2–15 (Table 1). The identified metabolites included four amines and 10 derivatives of amino acids *viz.* lysine, iso/leucine, phenylalanine, and glutamine, in which decarboxylation (-CO_2_, 44 amu), demethylation (-CH_2_, 14 amu), deglycosilation, deamination (-NH_2_, 17 amu) and dehydration (-H_2_O, 18 amu) were the major fragmentation pathways matching the reported literature, GNPS library, and HMDB and MassBank databases. All identified metabolites were common between both fruits except for peaks 14 and 15 (Table 1), which were detected in *L. siceraria* at [M+H]^+^ 267.1337, C_13_H_19_N_2_O_4_^+^ and [M+H]^+^ 245.1859, C_12_H_25_N_2_O_3_^+^, identified as N-(methoxybenzyl)glutamine, which is previously reported in Cucurbitaceae [15] and Iso/Leucyl-Iso/Leucine dipeptide, respectively.

#### 3.1.2. Organic Acids

6 Organic acids were identified in peaks 17–22 (Table 1) and grouped in cluster F in negative MN (Figure 2) based on decarboxylation and dehydration shared fragments. For example, homocitric acid in peak 18 (Table 1) was detected in *L. siceraria* for the first time at [M-H]^−^, C_7_H_9_O_7_^−^ and fragmented into *m*/*z* 161, C_6_H_9_O_5_^−^ (-COO, 44 amu), *m*/*z* 125, C_6_H_5_O_3_^−^ [M-H-CO_2_-2H_2_O], and *m*/*z* 81, C_5_H_5_O^−^ [M-H-2CO_2_-2H_2_O].

#### 3.1.3. Alkaloids

Aside from previous studies that reported the presence of alkaloids in cucumber at moderate levels [3,16,17], peak 16 at [M+H]^+^ 247.1446, C_14_H_19_N_2_O_2_^+^ appeared as a singleton in positive MN (Appendix A) at Rt. 11.4 min. (Table 1) and was identified as N,N,N-trimethyltryptophan betaine, known as lenticin or hypaphorin alkaloid, which is reported herein for the first time in cucumber fruit. Identification was based on the presence of diagnostic daughter ions at *m*/*z* 188, C_11_H_10_NO_2_^+^, post the loss of N-trimethyl moiety, followed by decarboxylation at *m*/*z* 144 C_10_H_10_N^+^ or dehydration at *m*/*z* 170, C_11_H_8_NO^+^. The latter proceeded into ring cleavage and demethylation at *m*/*z* 146, C_9_H_8_NO^+^ and *m*/*z* 122 C_7_H_8_NO^+^, as explained in Appendix A, matching the reported literature [18] and HMDB spectrum (https://hmdb.ca/spectra/ms_ms/2947783, accessed on 11 January 2023). Lenticin is distributed in various vegetables and known to exhibit cardioprotective and neurological effects, and thus could be correlated with cucumbers’ cardioprotective reported effect for the first time using such a metabolomics approach [19]. It is believed that cucumber may encompass several other alkaloids, but a special extract targeting method with solvents of lower polarity is needed to enhance their detection.

#### 3.1.4. Phenolics and Cinnamic Acid Derivatives

In total, 15 glycosylated and/or acylated derivatives of phenolic and cinnamic acids were eluted at Rt. 8–16 min. in peaks 23–38 (Table 1). MSn ions for cinnamic acid derivatives were evidenced at *m*/*z* 179, C_9_H_7_O_4_^−^, *m*/*z* 161, C_9_H_5_O_3_^−^ and *m*/*z* 135, C_8_H_7_O_2_^−^ for caffeoyl, *m*/*z* 193, C_10_H_9_O_4_^−^ and *m*/*z* 175, C_10_H_7_O_3_^−^ for feruloyl, *m*/*z* 223, C_11_H_11_O_5_^−^ and *m*/*z* 205, C_11_H_9_O_4_^−^ for sinapoyl moieties (Table 1). While sinapic acid derivatives were detected in cucumber only, bottle gourd extract was exclusively enriched with conjugates of phenyl and caffeoyl or feruloyl glycosides (Appendix A) that appeared in negative MN grouped in cluster J (Figure 2) and were identified in peaks 28, 29, 31–33 and 35–38 (Table 1). Such conjugates are reported herein for the first time in the genus *Lagenaria* based on the diagnostic peaks at *m*/*z* 123, C_7_H_7_O_2_^–^ and *m*/*z* 105, C_7_H_5_O^−^ for hydroxymethylphenyl fragments or *m*/*z* 137, C_7_H_5_O_3_^–^ and *m*/*z* 93, C_6_H_5_O^−^ for hydroxybenzoic acid fragments, or *m*/*z* 139, C_7_H_7_O_3_^−^ and *m*/*z* 121, C_7_H_5_O_2_^−^ for dihydroxymethoxybenzene fragments (Appendix A). Hexosyl moieties in peaks 32, 37, and 38 (Table 1) were additionally acylated with malonic acid (C_3_H_2_O_3_^−^, 86 amu), which is known to improve the biological effects of phenolics [20].

For example, peak 35 detected in bottle gourd at [M-H]^−^ 461.1449, C_23_H_25_O_10_^−^ was identified as hydroxymethylphenyl-*O*-feruloyl-*O*-hexoside fragmented into *m*/*z* 337, C_16_H_17_O_8_^−^ post losing hydroxymethylphenyl moiety, *m*/*z* 193, C_10_H_9_O_4_^−^ for ferulic acid and its dehydrated and demethylated fragments at *m*/*z* 175, C_10_H_7_O_3_^−^ and *m*/*z* 160, respectively, whereas ions at *m*/*z* 123, C_7_H_7_O_2_^−^ and *m*/*z* 105, C_7_H_5_O^−^ were detected for hydroxymethlphenyl ion and its dehydrated form, respectively (Appendix A). The node of peak 35 in negative MN was directly connected to molecular ion [M-H]^−^ 547.1452 in peak 38 at C_26_H_27_O_13_^−^ having an extra C_3_H_2_O_3_^−^ (malonyl, 86 amu), and thus identified as hydroxymethylphenyl-*O*-feruloyl-*O*-malonylhexoside, which, in turn, was connected to [M-H]^−^ 709.1985, C_32_H_37_O_18_^−^ having an additional C_6_H_10_O_5_^−^ (hexosyl, 162 amu), and hence identified as hydroxymethylphenyl-*O*-hexosylferuloyl-*O*-malonylhexoside, aided by the necessary MSn ions and their predicted formulae to confirm their identification (Table 1). Little information is available on the bioactivities of such acylated conjugates and how they contribute to *L. siceraria* health effects; thus, further studies are required to explore their potential pharmacological effect.

#### 3.1.5. Sesquiterpenes

The family Cucurbitaceae is known to accumulate di-/sesqui-/and triterpenes, to which various bioactivities are attributed [4,21] nevertheless, sesquiterpenes are the least reported in the literature. Cluster H in negative MN consists of five metabolites (Figure 2), from which peak 39 was detected in both fruit extracts and was identified as cymaroside A at [M-H]^−^ 443.1918, C_21_H_31_O_10_^−^ (Table 1). Cynaroside A is sesquiterpene lactone-*O*-hexoside, abundant in artichoke, that undergoes deglycosilation at *m*/*z* 281, C_15_H_21_O_5_^−^, decarboxylation at *m*/*z* 237, C_14_H_21_O_3_, followed by sequential demethylation, dehydration and ring cleavage fragmentation processes to yield several diagnostic daughter ions, as explained in detail in Appendix A, in accordance with databases and the other literature [22]. Despite being previously detected/isolated from other green vegetables [23], this is the first report of its presence in F. Cucurbitaceae, and whether these fruits could serve as source of that sesquiterpene, similar to artichoke, should be considered. The other four metabolites clustered with cymaroside A sesquiterpene shared ions corresponding to lactone ring cleavage followed by sequential demethylation and dehydration at *m*/*z* 119, *m*/*z* 101, *m*/*z* 89, *m*/*z* 71, *m*/*z* 59 (Appendix A), but could not be completely identified.

#### 3.1.6. Alkyl Glycosides

From cluster D in negative MN (Figure 2), four metabolites were detected exclusively in cucumber as formate adducts in peaks 44–47 that belong to the alkyl glycosides class (Table 1). Alkyl glycosides are formed of fatty alcohols glycosylated with sugars. In detail, peak 44, [M-H]^−^ detected at *m*/*z* 367.1608, C_15_H_27_O_10_^−^ which was identified as butanol-*O*-pentosyl-hexoside and yielded fragment ions at *m*/*z* 235, C_10_H_19_O_6_^−^ and *m*/*z* 73, post sequential loss of both sugars followed by demethylation at *m*/*z* 59, while MSn ions at *m*/*z* 161, 113, 101, 71 belonged to hexosyl fragments (Appendix A) in good accordance with the literature [24] and databases. Similarly, peaks 45 and 46 were identified as benzyl-*O*-pentosyl-hexoside and hexanol-*O*-pentosyl-hexoside, respectively. Such compounds were previously detected in several fruits as in apples, anise, cumin, and others [15,25,26,27]; however, according to our knowledge, this is the first report of their presence in Cucurbitaceae. The exact bioactivities of such class of compounds in these fruits have yet to be studied.

#### 3.1.7. Flavonoids

Previous studies reported the presence of isoflavones and acylated/methoxylated apigenin and luteolin-*O*/*C*-glycosides in cucumber [28,29], while flavonoid-*O*/*C*-glycosides were reported in both species [3,4,30]. In this study, 24 flavonoids were detected and tentatively identified in both species, including flavones, flavonols, and isoflavones. *C*-glycosides were recognized by MSn ions corresponding to sugar ring cleavage, unlike *O*-glycosides, which showed intact release of the dehydrated sugar. Aglycones were differentiated based on their typical RDA fragments [31]. Identification was guided by GNPS-based networking, which allowed extrapolating peaks’ annotations to unknown compounds, as observed in clusters B and C, in positive and negative MNs, respectively (Appendix A and Figure 2).

*Acylated flavonoids* were detected in peaks 52, 54, 61, 62, 66–68, and 71 (Table 1), in which cucumber was more abundant in aromatic, i.e., feruloyl (C_10_H_8_O_3_, 176 amu) or coumaroyl (C_9_H_6_O_2_, 146 amu), acylated flavones, whereas bottle gourd was enriched with flavone and flavonols acylated with aliphatic, i.e., malonyl (C_3_H_2_O_3_, 86 amu) or acetyl (C_2_H_2_O, 42 amu), moieties. These differential metabolomics results could aid in annotating acetyl transferase enzyme by correlating metabolite data with gene expression data. For example, peak 54 was identified as iso/vitexin-*O*-couamroylhexoside, which was previously reported in cucumber [28]. The precursor ion detected at [M+H]^+^ 741.2041, C_36_H_37_O_17_^+^ showed fragments at *m*/*z* 595, C_27_H_31_O_15_^+^ [loss of coumaroyl, 146 amu], *m*/*z* 433 for iso/vitexin, C_21_H_21_O_10_^+^ upon loss of *O*-hexosyl moiety, followed by MSn ions for *C*-hexosyl ring cleavage and aglycone RDA at *m*/*z* 415, *m*/*z* 397, *m*/*z* 313 and *m*/*z* 283 (Appendix A).

*Methoxylated flavonoids* were detected abundantly in bottle gourd fruit as in peaks 55, 58, 60, 62, 63, 64, 66–69, and 71 (Table 1), identified based on the neutral loss of one or more methyl (-CH_2_, 14 amu) / methoxyl (-OCH_2_, 30 amu) moieties. For example, peak 62 was identified as isorhamnetin-*O*-malonylhexoside at [M+H]^+^ 565.1186, C_25_H_25_O_15_^+^ and was reported for the first time in both species (Appendix A). The precursor ion yielded fragments at *m*/*z* 479 [M+H-malonyl]^+^, *m*/*z* 317, C_16_H_13_O_7_^+^ for isorhamnetin post deglycosylation followed by demethylation at *m*/*z* 303, C_15_H_11_O_7_, dehydration at *m*/*z* 285, C_15_H_9_O_6_^+^ and finally RDA fragments at *m*/*z* 229, *m*/*z* 151 and *m*/*z* 127 [32]. To our knowledge, this is the first report of di/methoxylated flavonoid-*O*-acylated hexoside in *L. siceraria,* as in peaks 64, 66–69, and 71 (Table 1).

Such a mixture of flavonoids promotes the various health effects of both species, since acylation and methoxylation are known to impart structural and functional modifications that improve flavonoids’ bioactivity by increasing their lipophilicity and intracellular bioavailability, thus exhibiting better receptor/ligand binding compared to their non-acylated/methoxylated analogues, as observed in cancer chemoprevention and anti-acetylcholine esterase activities [20,33,34,35].

#### 3.1.8. Pterocarpans

Pterocarpans are derivatives of isoflavonoids de novo biosynthesized as a response to stress, mainly detected in legumes [36,37]. Herein, three pterocarpans were detected in cucumber grouped with flavonoids in negative MN cluster C (Figure 2) and identified in peaks 72–74 (Table 1). Identification was based on the diagnostic fragmentation pattern, showing sequential loss of two methyl groups (−2 × CH_2_, 14 amu) followed by decarbonylation (-CO, 28 amu) confirmed by generated formulae [38]. For example, peak 72 in Table 1 showed the precursor ion at [M-H]^−^ 475.124, C_23_H_23_O_11_^−^ that yielded MSn ions at *m*/*z* 313, C_17_H_13_O_6_^−^ [M-H-C_6_H_10_O_5_, hexosyl 162 amu], followed by demethylation at *m*/*z* 298 and *m*/*z* 283, C_15_H_7_O_6_^−^, then loss of carbonyl at *m*/*z* 255, C_14_H_7_O_5_^−^ and was thus identified as hedysarimpterocarpene A-*O*-hexoside (HPA-*O*-hexoside) in accordance with the reported literature [38] (Appendix A). Peak 72 node in cluster C was directly connected to [M-H]^−^ 517.1345, C_25_H_25_O_12_^−^, having an additional 42 amu (-C_2_H_2_O^−^) and sharing the same fragments at *m*/*z* 313, *m*/*z* 298 and *m*/*z* 255, and was thus identified as hedysarimpterocarpene A-*O*-acetylhexoside (HPA-*O*-acetylhexoside). Similarly, peak 74 at [M-H]^−^ 313.0715, C_17_H_13_O_6_^−^ was identified as hedysarimpterocarpene (HPA) (Table 1). This is the first report of pterocarpans in Cucurbitaceae; further studies are required to confirm their biosynthesis in plants other than legumes and present them as potential sources of pterocarpans.

#### 3.1.9. Lignans

Three lignans belonging to the dibenzylbutanediol subclass were grouped in cluster A in positive MN as detected in bottle gourd for the first time at [M+H]^+^ 379.1747, [M+H]^+^ 363.1795 and [M+H]^+^ 347.1829 in peaks 75–77 (Table 1), identified as pentahydroxy-dimethoxylignan, known as carinol and secoisolariciresinol, previously reported in Cucurbitaceae [39], and trihydroxy-dimethoxylignan, respectively. Elemental composition of peaks 76, C_20_H_27_O_6_^+^ and 77, C_20_H_27_O_5_^+^ showed one and two hydroxyl moieties fewer than peak 75, C_20_H_27_O_7_^+^, respectively. Identification was based on diagnostic fragments corresponding to sequential dehydration, demethylation, and demethoxylation prior to and/or after cleavage of the bis(benzylbutanediol) bonding (Appendix A) sharing MSn ions at *m*/*z* 137, C_8_H_9_O_2_^+^, *m*/*z* 121, C_8_H_9_O^+^ *m*/*z* 107, C_7_H_7_O^+^ and *m*/*z* 93, C_7_H_9_^+^, in accordance with references and databases [40].

#### 3.1.10. Saponins

Previously, cucumber was reported for the presence of triterpenoid saponins [41]. In this study, five saponins in peaks 78–82 were detected in cucumber fruit (Table 1). Peak 82 detected in positive ionization mode at [M+H]^+^ 943.5275, C_48_H_79_O_18_^+^ was identified as soyasaponin I based on fragments at *m*/*z* 797, [M+H-rhm], *m*/*z* 635, C_36_H_59_O_9_ [(M+H)-rhm-hex], 617, C_36_H_57_O_8_
^+^, *m*/*z* 599, C_36_H_55_O_7_^+^, *m*/*z* 581, C_36_H_53_O_6_^+^ and *m*/*z* 459, C_30_H_51_O_3_^+^ for soyasapogenol B post loss of glucuronide moiety (C_6_H_8_O_6_,176 amu) followed by dehydration, demethylation and RDA fragmentation of sapogenin at *m*/*z* 441, C_30_H_49_O_2_^+^, *m*/*z* 423, C_30_H_47_O^+^, *m*/*z* 405, C_30_H_45_^+^, *m*/*z* 383, C_27_H_43_O^+^ and other fragments (Appendix A) matching references and databases [42]. On the other hand, all saponins were detected in negative ionization mode analysis as formate adducts (+46 amu, HCOOH), in which soyasaponin I showed few fragments indicative for the loss of hydroxymethyl, hydroxyl and ring A cleavage at *m*/*z* 397 and *m*/*z* 341, as predicted in HMDB spectra. In negative MN (Figure 2), cluster E consists of saponins correlated with soyasaponin I for their soyasapogenol B/E MSn ions at *m*/*z* 397 and *m*/*z* 341, in addition to other ions indicating terminal sugar loss or a fragment of hexose at *m*/*z* 113 (Table 1, Appendix A). From the aforementioned data, peaks 79 and 81 were annotated as soyasapogenol E-*O*-dihexosyl-*O*-glucuronide (soyasaponin Bd) and soyasapogenol B-*O*-rhamnosyl-*O*-pentosyl-*O*-hexosyl-*O*-glucuronide (melilotus saponin O1), respectively. The mixture of these saponins is reported in legumes [43], but this was the first time it was reported in cucumber. Other saponins maybe present in *L. siceraria* and *C. sativus* fruits that could be revealed upon using an extraction-targeting method instead of crude methanol extracts.

#### 3.1.11. Fatty Acids/Amides

In total, 16 fatty acids and one fatty acyl amide were annotated in peaks 83–99, equally distributed in both species (Table 1) and grouped in the major clusters in both negative and positive MNs, A and E, respectively (Figure 2 and Appendix A). This included mono-, di-, tri-, and tetrahydroxylated fatty acids, as evidenced by the loss of water (H_2_O,18 amu) and carboxylate (CO_2_, 44 amu) moieties. Six saturated fatty acids were detected in peaks 83, 84, 86, 87, 98, and 99. For example, peak 86—detected in both species— was identified as azelaic acid at [M-H]^−^ 187.0975, C_9_H_15_O_4_^−^ a dicarboxylic fatty acid, which showed fragment ions at *m*/*z* 169, C_9_H_13_O_3_^−^ [M-H-H_2_O], *m*/*z* 143, C_8_H_15_O_2_^−^ [M-H-CO_2_], *m*/*z* 125, C_8_H_13_O^−^ [M-H-H_2_O-CO_2_], *m*/*z* 97, C_6_H_9_O^−^, *m*/*z* 80, *m*/*z* 71, *m*/*z* 69 C_4_H_5_O^−^ and *m*/*z* 57, matching the literature [44]. Azelaic acid is incorporated into skin products for treatment of alopecia, as well as its reported cytotoxic and anti-inflammatory effects [44,45]. This could account for the popular usage of cucumber in skin preparations [3]. In contrast, 10 unsaturated fatty acids were detected in peaks 85, 88, and 90–97 (Table 1). Peak 85 was identified as octadecenamide at [M+H]^+^ 282.2796, C_18_H_36_NO^+^ showed fragment at *m*/*z* 265 C_18_H_33_O^+^ upon loss of ammonia (NH_3_, 17 amu). The herein identified fatty acids matched reported literature on Cucurbitaceae seed oil contents [15].

#### 3.1.12. Lysophospholipids

In negative MN (Figure 2), cluster B and cluster I grouped lysophospholipids and lysophosphatidic acids, respectively, from which peaks 100–107 were detected (Table 1). Metabolites detected at [M-H]^−^ 481.2203, C_21_H_38_O_10_P^−^, [M-H]^−^ 497.2156, C_21_H_38_O_11_P^−^, [M-H]^−^ 463.2100, C_21_H_36_O_9_P^−^ and [M-H]^−^ 431.2202, C_21_H_36_O_7_P^−^ are lysophophatidic acid derivatives, detected in cucumber only in peaks 100–102, and 104 (Table 1)

Overall, metabolome profiling of cucumber and bottle gourd based on GNPS-networking showed that both fruits share the abundance of amino acids, organic acids, flavones-*C*-glycosides, fatty acids and lysophospholipids in a comparable manner. By contrast, triterpenoid saponins, alkaloids, flavones-*C*-glycosides acylated with ferulic or coumaric acids, isoflavones alkyl glycosides and pterocarpans were exclusively detected in cucumber, and the latter two classes were reported for the first time in Cucurbitaceae. On the other hand, bottle gourd was further distinguished by the presence of dimethoxylated flavonoids acylated with malonic or acetic acids, lignans and conjugates of phenyl compounds with caffeic/ferulic acid-*O*-glycosides, all of which are reported for the first time in genus *Lagenaria*. Sesquiterpenes were more abundant in bottle gourd. except for cynaroside A, which was detected in both fruits for the first time as well. Successive extraction of fruits with solvents of different polarities is recommended to further reveal phytochemical classes that require targeting, such as alkaloids and saponins. The herein identified metabolites rationalized several of the reported pharmacological effects in both species, and further promoted them for new in vitro and in vivo medicinal research. Cucumber shared several phytochemicals that are characteristic to legumes (Fabaceae), e.g., soyasaponins, isoflavones, and pterocarpans, thus supporting the idea of an in-depth study of its biosynthetic pathways and gene expression data [37,43].

### 3.2. Aroma Profiling of L. siceraria and C. sativus Fruits Using SPME Coupled to GC-MS

SPME is well suited for the profiling of aroma for low-strength aroma food matrices, and also at low temperature for collection, compared to steam distillation, providing the true composition of volatile blends, [11] and this is the first time it has been reported in both species.

In this study, three independent replicates were analyzed under the same conditions via SPME-GC/MS and represented by total ion chromatograms (Figure 3A). 93 volatile constituents were detected in *L. siceraria* and *C. sativus* fruits and categorized into alcohols (11), aldehydes (16), ketones (14), fatty acids (3), oxides/ethers (7), S-containing compounds (1), monoterpenes (3), sesquiterpenes (5), esters (22), furans (1), and aliphatic (7) and aromatic (3) hydrocarbons (Appendix A). Both species showed comparable aroma composition, as seen in the relative percentile of each class in Figure 3B.

In *L. siceraria*, alcohols and ethers were the major volatile classes, detected at 23.34% and 23.52%, respectively, whereas aldehydes were the most abundant in *C. sativus* (25.63%), followed by ethers (23.93 %) (Figure 3B, Appendix A). Sesquiterpenes were detected in both fruits at low levels, i.e., 1.05% and 1.07% in bottle gourd and cucumber, respectively. Linalool is the major volatile constituent detected in both fruits at 18.76 % and 17.70 %, followed by safrole (11.4%, 10.5%), anethol (9.43%, 11.2%), and octanol (9.18%, 11.1%) in bottle gourd and cucumber, respectively.

Interestingly, hexenal, nonenal, and nonadienal (cucumber aldehyde) aldehydes that were reported to account for cucumber’s pleasant aroma [46] are herein detected in bottle gourd as well for the first time, at comparable levels except for 2,6-nonadienal (peak 19), which is more abundant in cucumber (Appendix A).

### 3.3. Multivariate Data Analysis of Volatiles Dataset Acquired from SPME-GC/MS

Although notable differences in volatile constituents were observed visually between both fruits, multivariate data analyses were employed to classify both fruits in an untargeted manner using PCA and OPLS modelling (Appendix A). Samples segregation was observed along PC1 and PC2 to account for more than 80% of the total variance (Appendix A). Supervised OPLS was further applied for its superiority in class separation [47] to obtain a model with good variance and prediction power (R^2^ = 0.88, Q^2^ = 0.82) (Appendix A). The loading plot and S-loading plots identified metabolites mediating for samples separation as denoted by Mol. ion/Rt and labelled with their peak numbers in accordance with Appendix A (Appendix A).

Examination of the PCA loading plot revealed that aldehydes were abundant in cucumber, represented by benzaldhyde (peak 14), octanal (peak 15), benzenacetaldehyde, (peak 17) and nonadienal (peak 19), in addition to anethol (peak 83), whereas bottle gourd was further enriched with ketones and esters, i.e., fenchone (peak 67) and methyl hexadecanoate (peak 58) (Appendix A, Appendix A).

Furthermore, OPLS score plot showed better discrimination of samples (Appendix A). The S-loading plot derived from OPLS model (Appendix A) confirmed aldehydes’ abundance in cucumber as discriminating metabolites, i.e., octanal (peak 15), benzenacetaldehyde (peak 17), and nonadienal (peak 19), versus anethol ether (peak 83) fenchone (peak 67), and methyl hexadecanoate (peak 58) in bottle gourd fruit.

The overall aroma of bottle gourd and cucumber is manifested by a mixture of characteristic volatiles. Linalool, that is, the major detected constituent in both fruits, is known to impart an orange oil-like aroma, whereas fenchone in bottle gourd possesses a pleasant camphor-like aroma and is incorporated in food as a perfumery flavor [48]. On the other hand, aldehydes in cucumber, i.e., nonadienal, nonenal, and hexenal are known to mediate for the pleasant characteristic aroma of fresh cucumber that is further enhanced upon chewing, while benzaldhyde has an almond-like odor, all of which are shared in bottle gourd as well at variable concentrations. [49,50,51]. In addition to their role in taste and odor, such volatiles are known to mediate for antimicrobial activity [51,52]. The study infers that bottle gourd possesses a consumer-pleasant aroma.

### 3.4. Unsupervised PCA Data Analysis of C. sativus and L. siceraria Primary Metabolites

To provide insight into *C. sativus* and *L. siceraria* fruits’ primary metabolome mediating for their nutritive value, GC–MS was employed, leading to the detection of 49 peaks. Chromatograms displayed a representative profile of *C. sativus* and *L. siceraria* fruits’ nutrient primary metabolites (Appendix A).

Metabolites were categorized into alcohols (6), amino acids (2), fatty acids/esters (12), nitrogenous compounds (1), organic acids (12), phenolic acids (1), steroids (2), sugars (9), sugar acids (1), and sugar alcohols (3), as shown in Table 2.

In *C. sativus*, sugars represented the most abundant class at ca. 63% of detected primary metabolites, mostly represented by monosaccharides at 61% and trace disaccharides at ca. 2%. Fructose, glucose, and galactose were the main identified monosaccharides, amounting to ca. 27, 19, and 14%, respectively, to account for the mild sweet taste of *C. sativus* (Table 2). The sugars in *L. siceraria* were detected at half that in *C. sativus* (*ca*. 33%), likewise represented by fructose, glucose, and galactose, though at lower levels than in *C. sativus* at 8–11%. Both samples encompassed trace levels of sugar acids and sugar alcohols (Table 2). Myo-inositol was the major detected sugar alcohol detected at 1%, which is in agreement with what was previously published for cucumber fruits [53].

12 Fatty acids/esters were identified in both fruit samples, i.e., cucumber and bottle gourd, amounting for ca. 15 and 36%, respectively (Table 2), mostly represented by saturated fatty acids detected at 13 and 29% in both fruit samples, respectively. The main identified fatty acids were palmitic, caprylic, and stearic acids (Table 2). At the cellular and tissue levels, palmitic acid performs a variety of important biological roles [54]. Caprylic acid is a medium-chain saturated fatty acid that exerts strong anti-atherosclerotic, anti-inflammatory, antifungal, and antibacterial effects [55,56]. Only three unsaturated fatty acids were identified, i.e., linoleic acid, oleic acid, and α-linolenic acid, amounting to 2 and 7% in cucumber and bottle gourd, respectively.

12 Organic acids were identified in both samples amounting to 13 and 17% in *C. sativus* and *L. siceraria*, respectively. Glutamic, malic, hydroxy butanoic, and lactic acids were the major identified organic acids in both fruits (Table 2). Organic acids are known to exert bactericidal activity acting as food acidulant preservatives [57].

Azelaic acid one of the organic acids identified, though at small levels, in both samples at 0.3%, and is well-reported for the treatment of a variety of skin conditions owing to its many skin-healing properties [58]. Whether *L. siceraria* has potential application in skin products comparable to that of C. sativus, with its well-reported anti-irritant, anti-acne, and skin-whitening properties has yet to be examined.

Amino acids were represented by two amino acids, viz. glycine and alanine, amounting to 2–3% of the total primary metabolites. Only one non-amino acid nitrogenous, i.e., urea, was identified in both fruits at 3–5% (Table 2).

### 3.5. Supervised Multivariate OPLS-DA Analysis of C. sativus and L. siceraria Primary Metabolites

(OPLS-DA) was further employed to confirm samples’ classification and primary metabolites’ marker determination obtained by GC-MS data-based PCA analysis. Both samples were modelled against each other (Appendix A), showing clear separation as shown in the derived score plot, with R^2^ = 0.99 (variance coverage) and a prediction goodness parameter Q^2^ = 0.93 (Appendix A). The corresponding S-plot (Appendix A) showed that cucumber fruits were rich in sugars, i.e., glucose, galactose, and fructose, as listed in Table 2, found at a level twice that found in *L. siceraria* fruit, which accounts for its better recognition as fresh food, owing to its better taste. By contrast, the bottle gourd fruits were rich in fatty acids (Appendix A), which confirmed the same markers obtained by PCA classification.

## 4. Conclusions

This study provided the first comparative metabolite profiling of cucumber and bottle gourd fruits via HR-UPLC/MS/MS based on GNPS-networking in both modes, allowing for annotation of 107 metabolites, i.e., amino acids (14), organic acids (6), phenolic/cinnamic acid derivatives (15), alkaloids (1), flavonoids (24), pterocarpans (3), alkyl glycosides (4), sesquiterpenes (5), saponins (5), lignans (3), fatty acids/amides (16), and lysophospholipids (8). Overall, both fruits share an abundance of amino acids, organic acids, flavones-*C*-glycosides, fatty acids and lysophospholipids in a comparable manner. However, triterpenoid saponins, alkaloids, flavones-*C*-glycosides acylated with ferulic or p-coumaric acids, alkyl glycosides, and pterocarpans were exclusively detected in cucumber, and the latter two classes were reported in this study for the first time to be present in Cucurbitaceae. On the other hand, bottle gourd was further distinguished by the presence of dimethoxylated flavonoids acylated with malonic or acetic acids, lignans, and conjugates of phenyl compounds with caffeic/ferulic acid-*O*-glycosides, all of which were reported for the first time in genus *Lagenaria.* Sesquiterpenes were more abundant in bottle gourd, except for cynaroside A, which was detected in both fruits for the first time. The herein-identified metabolites rationalized several of the reported pharmacological effects of both species and further promoted them for new in vitro and in vivo medicinal research. Several of the herein-detected phytochemicals require further bioactivity studies to study their potential health effects, e.g., pterocarpans, alkyl glycosides, and phenyl-cinnamic acids conjugates. Aroma profiling via SPME-GC/MS analysis detected 93 volatiles at comparable levels in both species, responsible for the pleasant aroma of bottle gourd, despite its enrichment with ketones and esters. 49 Silylated primary metabolites were detected in both species via GC/MS analysis at comparable levels, of which bottle gourd was further enriched with fatty acids, and cucumber with sugars.

## Figures and Tables

**Figure 1 foods-12-00771-f001:**
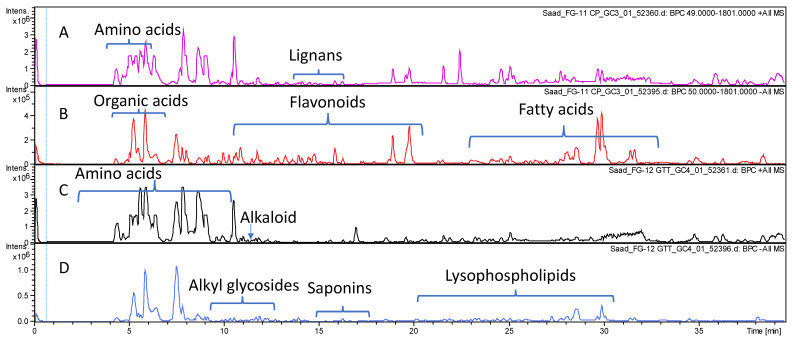
Base peak chromatograms of *Lagenaria siceraria* (**A**,**B**) and *Cucumis sativus* (**C**,**D**) crude fruit extracts as analyzed via HR-UPLC/MS/MS in both positive (**A**,**C**) and negative (**B**,**D**) ionization modes.

**Figure 2 foods-12-00771-f002:**
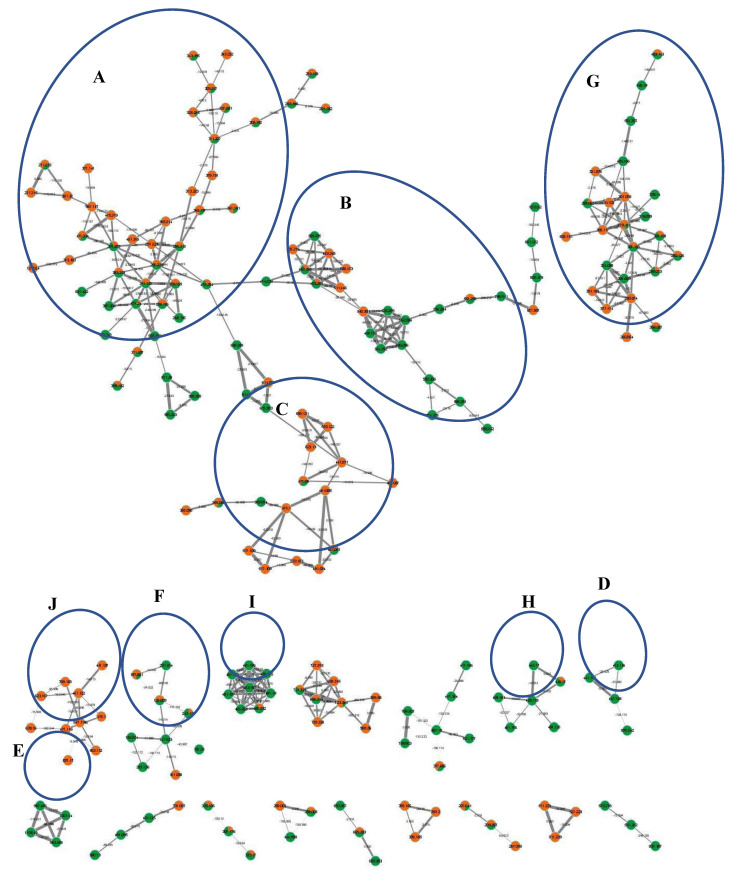
Full molecular networking created using MS/MS data in negative ionization mode for *L. siceraria* (bottle gourd) and *C. sativus* (cucumber) crude fruit extracts showing 351 nodes and 449 edges. All nodes are labelled with parent mass and edges are labelled with neutral loss values. The network is displayed as a pie chart with orange and green colors representing distribution of the precursor ion intensity in the bottle gourd and cucumber extracts, respectively. **Clusters annotation**: (**A**): fatty acids/amides, (**B**): lysophospholipids, (**C**): flavonoids and ptercarpans, (**D**): alkyl glycosides, (**E**): saponins, (**F**): organic acids, (**G**): amino acids derivatives, (**H**): sesquiterpenes, (**I**): lysophosphatidic acid derivatives, and (**J**): cinnamates.

**Figure 3 foods-12-00771-f003:**
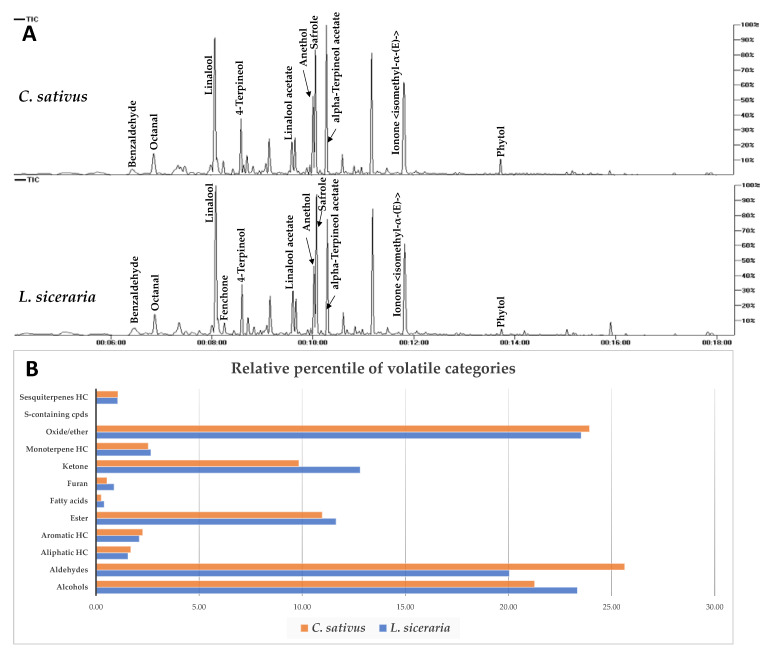
(**A**) Total ion chromatogram of *C. sativus* and *L. siceraria* aroma constituents as analyzed via SPME GC/MS. Assigned compounds names follow those shown in Appendix A. (**B**) Relative percentile of different volatile categories detected via SPME-GC/MS analysis in *C. sativus* and *L. siceraria* fruits.

**Table 1 foods-12-00771-t001:** Metabolome profiling of *Cucumis sativus* and *Lagenaria siceraria* crude fruit extracts via HR-UPLC/MS/MS in both negative and positive modes of analyses.

Peak	Identification	[M-H]-	[M+H]^+^	Elemental Composition	Error (ppm)	MSn ions	Rt (min)	*C. sativus*	*L. siceraria*
**Sugars**									
1	Disaccharide	377.0853		C_18_H_17_O_9_^−^	4	341, 215, 179	5.1	**+**	**+**
**Amino acids/amines**									
2	Lysine-*O*-hexosyl	307.1508		C_12_H_23_N_2_O_7_^−^	1	217, 187, 145, 127	4.6	**+**	**+**
3	N-trimethyl-lysine		189.1597	C_9_H_21_N_2_O_2_^+^	0.3	144, 130, 84	4.7	**+**	**+**
4	Pyroglutamic acid		130.499	C_5_H_8_NO_3_^+^	2	84	5	**+**	**+**
5	N-hexosyl-pyroglutamate	290.0895		C_11_H_16_NO_8_^−^	4.8	128	6.8	**+**	**+**
6	Pyrrolidine		72.0806	C_4_H_10_N^+^	1.9	56, 55	5.4	**+**	**+**
7	Nicotinamide		123.0555	C_6_H_7_N_2_O^+^	1.7	96, 80	6.6	**+**	**+**
8	Piperidine		86.0962	C_5_H_12_N^+^	2	70, 57, 56	6.7	**+**	**+**
9	Cyclo(leucylprolyl)		211.1437	C_11_H_19_N_2_O_2_^+^	1.7	192, 183, 154, 138, 114, 98, 86, 70	7	**+**	**+**
10	N-(deoxyhexosyl) phenylalanine		328.1395	C_15_H_22_NO_7_^+^	1.3	310, 292, 264, 246, 198, 178, 166, 132, 120	7.3	**+**	**+**
11	N-(deoxy-fructosyl)-iso/leucine	292.1402		C_12_H_22_NO_7_^−^	0	172, 130, 128, 101, 73	7.8	**+**	**+**
12	Pantothenic acid (Vit. B5)	218.1032		C_9_H_16_NO_5_^−^	1.1	146, 116, 99, 88	9.1	**+**	**+**
13	Methyl-pyroglutamic acid		144.066	C_6_H_10_NO_3_^+^	2	98, 84	9.5	**+**	**+**
14	N-(methoxybenzyl)glutamine		267.1337	C_13_H_19_N_2_O_4_^+^	0.7	250, 237, 232, 221, 211, 203, 166, 136, 121, 101, 86, 74	9.9	**-**	**+**
15	Iso/Leucyl-Iso/Leucine		245.1859	C_12_H_25_N_2_O_3_^+^	0.4	228, 210, 132, 120, 86, 69	11.3	**-**	**+**
**Alkaloids**									
16	N,N,N-Trimethyltryptophan betaine (Lenticin alkaloid)		247.1446	C_14_H_19_N_2_O_2_^+^	2	188, 170, 146, 144, 118, 60	11.4	**+**	**-**
**Organic acids**									
17	Shikimic acid	173.0455		C_7_H_9_O_5_^−^	0.3	137, 93	5.7	**-**	**+**
18	Homocitric acid	205.0352		C_7_H_9_O_7_^−^	0.8	161, 125, 117, 81, 73, 59	5.75	**-**	**+**
19	Malic acid	133.0139		C_4_H_5_O_5_^−^	2.6	115, 71	5.8	**+**	**+**
20	Citric acid	191.0195		C_6_H_7_O_7_^−^	1	173, 129	7.5	**-**	**+**
21	Citramalic acid	147.0304		C_5_H_7_O_5_^−^	3	129, 103, 87	7.9	**+**	**-**
22	Hydroxy-methylglutaric acid	161.0453		C_6_H_9_O_5_^−^	1.5	99, 57	8	**-**	**+**
**Phenolic and cinnamic acids derivatives**									
23	Dihydroxy-methoxybenzene-*O*-hexoside (Methoxycatechol-*O*-hexoside)	301.0928		C_13_H_17_O_8_^−^	0.1	139, 121, 93	7.9	**-**	**+**
24	Caffeoyl-*O*-hexoside	341.0878		C_15_H_17_O_9_^−^	0.1	179	10.1		**+**
25	Sinapic acid		225.0763	C_11_H_13_O_5_+	2	207, 192, 175, 157, 123, 119	11.6	**+**	
26	Sinapoyl-*O*-hexoside	385.1138		C_17_H_21_O_10_^−^	0.6	223, 179, 164	11.7	**+**	
27	Feruloyl-*O*-hexoside	355.1035		C_16_H_19_O_9_^−^	1	265, 235,193, 175, 161, 149, 134	12.2	**+**	**+**
28	Dihydroxy-methoxybenzene -*O*-hexosylferuloyl-*O*-hexoside	639.1918		C_29_H_35_O_16_^−^	1.9	499, 463, 445, 431, 337, 193, 175, 139, 121,	12.9	**-**	**+**
29	Hydroxymethylphenyl -*O*-hexosylferuloyl-*O*-hexoside	623.1981		C_29_H_35_O_15_^−^	0	499, 461, 337, 193, 175, 161, 123, 105	13.1	**-**	**+**
30	Caffeic acid	179.035		C_9_H_7_O^−^	1	135, 107	10	**-**	**+**
31	Hydroxymethylphenyl -*O*-Caffeoyl-*O*-hexoside	447.1295		C_22_H_23_O_10_^−^	0.5	323, 179, 161, 135, 123, 105	13.4	**-**	**+**
32	Hydroxymethylphenyl -*O*-hexosylferuloyl-*O*-malonylhexoside	709.1985		C_32_H_37_O_18_^−^	0	665, 623, 499, 337, 193, 175, 119, 113	14.3	**-**	**+**
33	Dihydroxy-methoxybenzene -*O*-Feruloyl-*O*-hexoside	477.1398		C_23_H_25_O_11_^−^	1	337, 314, 193, 175, 139, 121, 103, 73	14.5	**-**	**+**
34	Ferulic acid		195.0648	C_10_H_11_O_4_^+^	0.4	177, 163	14.6	**-**	**+**
35	Hydroxymethylphenyl -*O*-Feruloyl-*O*-hexoside	461.1449		C_23_H_25_O_10_^−^	0.1	337, 193, 175, 160, 123, 105	14.7	**-**	**+**
36	Hydroxybenzoic acid -*O*-Feruloyl-*O*-hexoside	475.1237		C_23_H_23_O_11_^−^	1.9	337, 193, 175, 161, 137, 93	14.9	**-**	**+**
37	Dihydroxy-methoxybenzene -*O*-Feruloyl-*O*-malonylhexoside	563.1401		C_26_H_27_O_14_^−^	1	337, 193, 175, 121	15.7	**-**	**+**
38	Hydroxymethylphenyl -*O*-Feruloyl-*O*-malonylhexoside	547.1452		C_26_H_27_O_13_^−^	1	503, 461, 337, 193, 175, 160	16.1	**-**	**+**
**Sesquiterpenes**									
39	Cynaroside A	443.1918		C_21_H_31_O_10_^−^	0.7	281, 237, 219, 189, 161, 131, 119, 113, 89, 71, 59	10	**+**	**+**
40	Unknown	429.1394		C_19_H_25_O_11_^−^	1.8	205, 119, 113, 89, 87, 73, 71	13	**+**	**-**
41	Unknown	407.1553		C_17_H_27_O_11_^−^	1.3	243, 183, 119, 101, 89, 87, 73, 59		**+**	**-**
42	Unknown	409.1716		C_17_H_29_O_11_^−^	0.2	269, 225, 89, 87, 59	14.2	**+**	**-**
43	Unknown	461.2021		C_21_H_33_O_11_^−^	1.6	343, 161, 153, 101, 87, 73, 71	16.5	**+**	**-**
**Alkyl glycosides**									
44	Butanol-*O*-pentosyl-hexoside	367.1608		C_15_H_27_O_10_^−^	0.5	235, 203, 161, 131, 113, 101, 85, 73 71, 59	10.2	**+**	**-**
45	Benzyl-*O*-pentosyl-hexoside	401.1448		C_18_H_25_O_10_^−^	1.8	269, 161, 131, 113, 101, 85, 71	11.1	**+**	**-**
46	Hexanol-*O*-pentosyl-hexoside	395.1928		C_17_H_31_O_10_^−^	1.4	263, 161, 131, 101, 71	13.2	**+**	**-**
47	Unknown	529.2643		C_26_H_41_O_11_^−^	2	397, 347, 89	15.1	+	-
**Flavonoids**									
48	Isovitexin-*O*-hexoside (Saponarin)		595.167	C_27_H_31_O_15_+	0.1	577, 475, 433, 397, 313, 283	11.2	**+**	**+**
49	Iso/orientin	447.0933		C_21_H_19_O_11_^−^	1	429, 369, 357, 327, 297, 285	12.1	**-**	**+**
50	Iso/vitexin	431.0974		C_21_H_19_O_10_^−^	2.2	341, 311, 283, 269	13.3	**+**	**+**
51	Quercetin-*O*-hexoside	463.0878		C_21_H_19_O_12_^−^	0.8	300, 271, 255, 179, 151	13.5	**-**	**+**
52	Iso/vitexin-*O*-feruloylhexoside		771.2141	C_37_H_39_O_18_^+^	1.3	753, 595, 433, 415, 397, 337, 313, 283, 271, 195, 177, 145	13.5	**+**	**-**
53	Kaempferol -*O*-(2′-rhamnosyl)-hexoside		595.1652	C_27_H_31_O_15_^+^	0.9	327, 285, 151	13.58	**-**	**+**
54	Iso/vitexin-*O*-coumaroylhexoside		741.2041	C_36_H_37_O_17_^+^	2.1	595, 579, 433, 415, 397, 337, 313, 283, 271, 165, 147, 145	13.6	**+**	**-**
55	Isorhamentin-*O*-rutinoside		625.1756	C_28_H_33_O_16_^+^	1	479, 317, 302, 286,	13.65	**-**	**+**
56	Quercetin		303.0493	C_15_H_11_O_7_^+^	2	253, 121, 107	13.9		**+**
57	Kaempferol-*O*-hexoside	447.0927		C_21_H_19_O_1_1^−^	1.3	284	14.3	**+**	**+**
58	Isorhamnetin-*O*-hexoside	477.1031		C_22_H_21_O_12_^−^	1.5	314, 153	14.4	**+**	**+**
59	Luteolin		287.0549	C_15_H_11_O_6_^+^	0.3	255, 151, 133	14.5	**-**	**+**
60	Isorhamnetin		317.0653	C_16_H_13_O_7_^+^	0.7	302, 270	14.6	**-**	**+**
61	Luteolin-*O*-malonylhexoside		535.1083	C_24_H_23_O_14_^+^	0.1	431, 287, 231, 159	15.1	**-**	**+**
62	Isorhamnetin-*O*-malonylhexoside		565.1186	C_25_H_25_O_15_^+^	0.4	529, 479, 317, 302, 285, 245, 127	15.2	**+**	**+**
63	Chryseriol-*O*-hexoside		463.1233	C_22_H_23_O_11_^+^	0.4	301, 286, 258, 227, 211, 153, 145, 85	17.2	**+**	**+**
64	Dimethoxy-dihydroxyflavone-*O*-hexoside	491.1189		C_23_H_23_O_12_^−^	1.3	329, 313, 299, 271	17.3	**-**	**+**
65	Methylapigenin-*O*-hexoside		447.1295	C_22_H_23_O_10_^+^	2.2	285, 270, 229, 184, 159, 119	17.7	**+**	**+**
66	Methoxy-trihydroxyflavone-*O*-malonylhexoside		549.1237	C_25_H_25_O_14_^+^	0.3	301, 286, 231, 159, 85	17.75	**-**	**+**
67	Dimethoxy-trihydroxyflavone-*O*-malonylhexoside		579.1344	C_26_H_27_O_15_^+^	0	331, 316, 299, 271, 184, 159	17.85	**-**	**+**
68	* Dimethoxy-dihydroxyflavone-*O*-acetylhexoside	533.1297		C_25_H_25_O_13_^−^	0.6	329, 314, 299, 255	17.9	**-**	**+**
69	* Dimethoxy-trihydroxyflavone-*O*-hexoside		507.1497	C_24_H_27_O_12_^+^	0	345, 330, 315, 159	18.3	**-**	**+**
70	Dihydroxy-methoxy-isoflavone	283.061		C_16_H_11_O_5_^−^	0.6	268, 239, 211	18.7	**+**	**-**
71	* Dimethoxy-dihydroxyflavone-*O*-malonylhexoside		563.1396	C_26_H_27_O_14_^+^	0.1	315, 300, 184, 159	20	**-**	**+**
**Pterocarpans**									
72	Hedysarimpterocarpene A-*O*-hexoside (HPA-*O*-hexoside)	475.124		C_23_H_23_O_11_^−^	1.2	313, 298, 283, 255, 225	19.5	**-**	**+**
73	Hedysarimpterocarpene A-*O*-acetylhexoside (HPA-*O*-acetylhexoside)	517.1345		C_25_H_25_O_12_^−^	1.3	313, 298, 283, 255	19.8	**-**	**+**
74	Hedysarimpterocarpene A (HPA)	313.0715		C_17_H_13_O_6_^−^	0.7	298, 283, 270	20.3	**-**	**+**
**Lignans**									
75	Pentahydroxy-dimethoxylignan (Carinol)		379.1747	C_20_H_27_O_7_^+^	1.1	361, 347, 343, 315, 297, 285, 269, 253, 251, 225, 207, 197, 181, 137, 119, 105	13.1	**-**	**+**
76	Secoisolariciresinol		363.1795	C_20_H_27_O_6_^+^	1.9	327, 299, 281, 269, 253, 237, 223, 209, 195, 181, 143, 137, 121, 107, 103, 93	17.5	**-**	**+**
77	Trihydroxy-dimethoxylignan		347.1829	C_20_H_27_O_5_^+^	5	329, 311, 283, 255, 237, 207, 195, 183, 163, 137, 123, 107, 105	20.5	**-**	**+**
**Saponins**									
78	Unknown *O*-pentosyl-*O*-hexoside saponin derivative	917.5112		C_46_H_77_O_18_^−^	0.3	785, 397, 179	16.7	**+**	**-**
79	* Soyasapogenol E-*O*-dihexosyl-*O*-glucuronide (Soyasaponin Bd)	955.4889		C_48_H_75_O_19_^−^	2	397, 113	17.3	**+**	**-**
80	Unknown O-pentosyl saponin derivative	1059.5385		C_52_H_83_O_22_^−^	0.3	927, 397, 113	17.8	**+**	**-**
81	* Soyasapogenol B-*O*-rhamnosyl-*O*-pentosyl-*O*-hexosyl-*O*-glucuronide (Melilotussaponin O1)	1073.5534		C_53_H_85_O_22_^−^	0.4	927, 397, 341	18	**+**	**-**
82	Soyasaponin I	941.5107		C_48_H_77_O_18_	0.9	923, 397, 341, 113	19.9	**+**	**-**
			943.5275	C_48_H_79_O_18_+	1.5	797, 635, 617, 599, 581, 459, 441, 423, 405, 383			
**Fatty acids/amides**									
83	Trihydoxy-butanoic acid	135.0295		C_4_H_7_O_5_^−^	1.8	89, 75, 59	5.5	**+**	**-**
84	Undecane-tricarboxylic acid	287.15		C_14_H_23_O_6_^−^	0	269, 227, 209, 199, 59	14.8	**+**	**-**
85	Tetrahydroxy-octadecynoic acid	343.2124		C_18_H_31_O_6_^−^	0.6	325, 307, 289, 273, 229, 209, 171, 135, 83	15.5	**+**	**+**
86	Azelaic acid	187.0975		C_9_H_15_O_4_^−^	0.2	169, 143, 125, 97, 57	16.3	**+**	**+**
87	Octanedicarboxylic acid	201.1132		C_10_H_17_O_4_^−^	0.3	183, 139, 111, 57	18.1	**+**	**-**
88	Trihydroxy-octadecadienoic acid	327.2176		C_18_H_31_O_5_^−^	0.2	291, 211, 183	18.8	**+**	**-**
89	Octadecenamide		282.2796	C_18_H_36_NO^+^	1.7	265	18.9	**+**	**+**
90	Trihydroxyoctadecenoic acid	329.2332		C_18_H_33_O_5_^−^	0.6	311, 293, 229, 211, 171	19.7	**+**	**+**
91	Unsautrated fatty acid		275.2011	C_18_H_27_O_2_^+^	2.1	251, 219,185, 147, 133, 119, 105, 91, 81, 79, 67, 55	22	**+**	**+**
92	Dihydroxyoctadecadienoic acid	311.2227		C_18_H_31_O_4_^−^	0.2	293, 275, 199	25.2	**+**	**+**
93	Unsaturated fatty acid		277.2161	C_18_H_29_O_2_^+^	0.4	207, 133, 107, 93, 79, 67	25.4	**+**	**+**
94	Octadecatrienoic acid		279.2318	C_18_H_31_O_2_^+^	0.2	255, 149, 135, 107, 81,69	26	**+**	**+**
95	Linolenic acid derivative	559.3119		C_28_H_47_O_11_^−^	0.8	277	26.7	**+**	**-**
96	* Stearidonic acid	275.2017		C_18_H_27_O_2_^−^	0.2		28	**+**	**+**
97	Oxo-octadecatrienoic acid		293.2111	C_18_H_29_O_3_^+^	0	275, 257, 133, 107, 93, 81, 67	28.5	**+**	**+**
98	* Oxo-octadecanoic acid	297.2434		C_18_H_33_O_3_^−^	0.3	279, 171, 155	30.9	**+**	**+**
99	Hydroxy-tetracosanoic acid	383.3529		C_24_H_47_O_3_^−^	0.3	365, 309, 112	32.7	**+**	**+**
**Lysophospholipids**									
100	Lysophosphatidic acid derivative	481.2203		C_21_H_38_O_10_P^−^	1	255, 171, 153, 79	19.2	**+**	**-**
101	Lysophosphatidic acid derivative	497.2156		C_21_H_38_O_11_P^−^	0.3	171, 153, 79	19.8	**+**	**-**
102	Lysophosphatidic acid derivative	463.21		C_21_H_36_O_9_P^−^	0.5	335, 171, 153, 79	22.6	**+**	**-**
103	Palmitoyl-sn-glycero-phosphoglycerol	483.2725		C_22_H_44_O_9_P^−^	0.6	255	23	**-**	**+**
104	Hexosyl LPE 16:0	614.3308		C_27_H_53_NO_12_P^−^	0.4	452, 255	25.8	**+**	**+**
105	Palmitoyl-GPE	452.278		C_21_H_43_NO_7_P^−^	0.6	255, 196	27.7	**+**	**+**
106	LPE 18:2	476.278		C_23_H_43_NO_7_P^−^	0.6	280, 79	28.8	**+**	**+**
107	Lysophosphatidic acid derivative	431.2202		C_21_H_36_O_7_P^−^	0.3	277, 153, 79	30	**+**	**-**

* Metabolites identified based on GNPS networking via clustering with known compounds, (**+**) and (**-**) indicate presence and absence of a metabolite, respectively.

**Table 2 foods-12-00771-t002:** Primary metabolite analyzed using GC-MS of different *C. sativus* and *L. siceraria* with results expressed as relative percentile of the total peak area (n = 3).

Peak	Rt (min)	KI	Metabolite	M. formula	*C. sativus*	*L. siceraria*
**Alcohols**						
1	5.31	987	Ethylene glycol, 2TMS derivative	C_8_H_22_O_2_Si_2_	1.39 ± 0.37	2.44 ± 1.19
2	5.60	1003	Propylene glycol, 2TMS derivative	C_9_H_24_O_2_Si_2_	0.11 ± 0.03	0.18 ± 0.08
4	6.62	1061	1,3 Propanediol di-TMS	C_9_H_24_O_2_Si_2_	0.21 ± 0.06	0.40 ± 0.17
7	8.23	1151	1,4-Butanediol, 2TMS derivative	C_10_H_26_O_2_Si_2_	0.12 ± 0.03	0.18 ± 0.10
11	9.89	1251	Triethylene glycol, 2TMS derivative	C_12_H_30_O_4_Si_2_	0.49 ± 0.14	0.89 ± 0.39
13	10.41	1285	Glycerol, 3TMS derivative	C_12_H_32_O_3_Si_3_	0.85 ± 0.20	0.86 ± 0.30
			**Total alcohols**		**3.18 ± 0.82**	**4.95 ± 2.23**
**Amino acids**						
19	12.19	1401	Glycine, 3TMS derivative	C_11_H_29_NO_2_Si_3_	1.64 ± 0.27	2.41 ± 0.98
20	12.63	1434	β-Alanine, 3TMS derivative	C_12_H_31_NO_2_Si_3_	0.37 ± 0.08	0.69 ± 0.29
			**Total amino acids**		**2.01 ± 0.35**	**3.10** ± **1.28**
**Fatty acids/esters**						
18	11.57	1360	Caprylic acid n-butyl ester	C_12_H_24_O_4_	5.70 ± 1.64	9.74 ± 4.49
33	18.77	1948	12-Methyltetradecanoic acid TMS ester	C_18_H_38_O_2_Si	0.38 ± 0.18	0.14 ± 0.09
35	19.76	2045	Palmitic Acid, TMS derivative	C_19_H_40_O_2_Si	3.18 ± 1.04	10.76 ± 10.50
37	20.72	2144	Margaric acid, TMS ester	C_20_H_42_O_2_Si	0.13 ± 0.03	0.35 ± 0.15
38	21.38	2214	Linoleic acid TMS ester	C_21_H_40_O_2_Si	0.40 ± 0.26	1.02 ± 0.94
39	21.42	2218	Oleic acid, TMS ester	C_20_H_40_O_2_Si	0.93 ± 0.38	3.34 ± 3.76
40	21.46	2223	α-Linolenic acid, TMS	C_21_H_38_O_2_Si	0.77 ± 0.56	2.53 ± 3.14
41	21.63	2242	Stearic acid, TMS derivative	C_21_H_44_O_2_Si	2.78 ± 0.71	6.51 ± 3.11
42	23.36	2440	Arachidic acid, TMS derivative	C_23_H_48_O_2_Si	0.04 ± 0.03	0.09 ± 0.03
43	24.62	2600	1-Monopalmitin 2TMS ether	C_25_H_54_O_4_Si_2_	0.35 ± 0.15	0.84 ± 0.84
46	26.08	2786	Glycerol monostearate, 2TMS derivative	C_27_H_58_O_4_Si_2_	0.26 ± 0.12	0.39 ± 0.27
47	26.30	2812	Sebacic acid, bis(2-ethylhexyl) ester	C_26_H_50_O_4_Si_2_	0.38 ± 0.11	0.66 ± 0.29
			**Total fatty acids/esters**		**15.32 ± 5.19**	**36.38 ± 27.62**
**Nitrogenous compounds**						
10	9.80	1245	Urea, 2TMS derivative	C_7_H_22_N_2_OSi_2_	1.31 ± 0.48	4.10 ± 1.98
			**Total nitrogenous compounds**		**1.31 ± 0.48**	**4.10 ± 1.98**
**Organic acids**						
3	6.44	1051	Methylmalonic acid (2TMS)	C_10_H_22_O_4_Si_2_	0.17 ± 0.06	0.28 ± 0.14
5	6.74	1068	Lactic acid, 2TMS derivative	C_9_H_22_O_3_Si_2_	1.43 ± 0.45	0.99 ± 0.38
6	7.01	1083	Glycolic acid, 2TMS derivative	C_8_H_20_O_3_Si_2_	0.22 ± 0.07	0.26 ± 0.09
9	9.74	1241	4-Hydroxybutanoic acid, 2TMS deriv.	C_10_H_24_O_3_Si_2_	1.77 ± 0.44	3.83 ± 1.81
12	9.94	1254	Benzoic acid, TBDMS derivative	C_13_H_20_O_2_Si	0.17 ± 0.05	0.21 ± 0.12
14	10.42	1286	Phosphoric acid, tris(trimethylsilyl) ester	C_9_H_24_O_4_PSi_3_	2.35 ± 0.85	2.22 ± 1.41
**Peak**	**Rt (min)**	**KI**	**Metabolite**	**M. formula**	** *C. sativus* **	** *L. siceraria* **
15	10.88	1315	L-Glutamic acid, N,N-di(3-methylbutyl)-, dimethyl ester	C_17_H_33_NO_4_	3.35 ± 0.93	5.92 ± 2.46
16	10.96	1320	Succinic acid (2TMS)	C_10_H_22_O_4_Si_2_	0.75 ± 0.21	1.21 ± 0.53
21	13.51	1500	Malic acid, 3TMS derivative	C_13_H_30_O_5_Si_3_	2.45 ± 0.35	1.00 ± 0.41
22	14.11	1545	3-Methylvaleric acid, TMS	C_9_H_20_O_2_Si	0.27 ± 0.09	0.47 ± 0.21
23	14.29	1559	Erythronic acid, tetrakis(trimethylsilyl) deriv.	C_16_H_40_O_5_Si_4_	0.06 ± 0.03	0.01 ± 0.01
25	17.19	1799	Azelaic acid, 2TMS derivative	C_15_H_32_O_4_Si_2_	0.26 ± 0.16	0.34 ± 0.13
			**Total organic acids**		**13.24 ± 3.69**	**16.74 ± 7.70**
**Phenolic acids**						
45	25.62	2727	3,5-Dimethoxymandelic acid, di-TMS	C_16_H_28_O_5_Si_2_	0.11 ± 0.04	0.20 ± 0.08
			**Total phenolic acids**		**0.11 ± 0.04**	**0.20 ± 0.08**
**Sterols**						
48	29.19	3180	Cholesterol, TMS derivative	C_30_H_54_OSi	0.37 ± 0.29	0.09 ± 0.06
49	31.11	3423	Lanost-7-en-3-ol, 9,11-epoxy-, acetate, (3β,11α)-	C_32_H_52_O_3_	0.08 ± 0.04	0.48 ± 0.50
			**Total sterols**		**0.45 ± 0.33**	**0.57 ± 0.57**
**Sugars**						
26	17.55	1832	D-Fructopyranose, 5TMS derivative	C_21_H_52_O_6_Si_5_	9.72 ± 2.30	4.71 ± 2.82
27	17.65	1842	D-Fructose, 5TMS derivative	C_21_H_52_O_6_Si_5_	13.84 ± 1.53	5.59 ± 5.52
28	17.89	1865	Arabinofuranose, 4TMS derivative	C_17_H_42_O_5_Si_4_	0.88 ± 0.24	0.90 ± 0.50
29	17.89	1865	1-Deoxyglucose 4TMS	C_18_H_44_O_5_Si_4_	0.14 ± 0.02	0.16 ± 0.09
30	18.01	1877	D-Mannose, 5TMS derivative	C_21_H_52_O_6_Si_5_	0.04 ± 0.03	0.11 ± 0.03
31	18.39	1912	D-Fructose, 5TMS derivative isomer	C_21_H_52_O_6_Si_5_	3.01 ± 0.77	1.94 ± 1.22
32	18.44	1917	Galactopyranose, 5TMS derivative	C_21_H_52_O_6_Si_5_	14.24 ± 2.05	7.58 ± 5.59
34	19.33	2001	β-D-Glucopyranose, 5TMS derivative	C_21_H_52_O_6_Si_5_	19.36 ± 2.45	11.36 ± 8.18
44	25.34	2691	Sucrose, 8TMS derivative	C_36_H_86_O_11_Si_8_	1.55 ± 0.28	0.16 ± 0.05
			**Total sugars**		**62.78 ± 9.68**	**32.51 ± 23.99**
**Sugar acids**						
17	11.29	1342	Glyceric acid, 3TMS derivative	C_12_H_30_O_4_Si_3_	0.17 ± 0.03	0.02 ± 0.02
			**Total sugar acids**		**0.17 ± 0.03**	**0.02 ± 0.02**
**Sugar alcohols**						
8	8.33	1156	1,4-Anhydro-d-galactitol TMS derivative	C_6_H_12_O_5_	0.12 ± 0.02	0.24 ± 0.12
24	16.35	1729	Arabinitol, 5TMS derivative	C_20_H_52_O_5_Si_5_	0.14 ± 0.08	0.06 ± 0.01
36	20.49	2121	Myo-inositol, 6TMS derivative	C_24_H_60_O_6_Si_6_	1.18 ± 0.19	1.14 ± 0.47
			**Total sugar alcohols**		**1.43 ± 0.29**	**1.44 ± 0.60**

## Data Availability

The data presented in this study are available in article and Appendix A.

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
