# Peer review of "How Does Lagenaria siceraria (Bottle Gourd) Metabolome Compare to Cucumis sativus (Cucumber) F. Cucurbitaceae? A Multiplex Approach of HR-UPLC/MS/MS and GC/MS Using Molecular Networking and Chemometrics"

_foods, 2023, doi:10.3390/foods12040771_

Round 1

Reviewer 1 Report

The manuscript investigates the chemical composition of cucumber and bottle gourd, in a comparative approach leading to the identification of several first time-reported metabolites and classes in both species and rationalizing better for their health effects and potential application as nutraceuticals in the future-based on such chemical profiling. For this, a multiplex approach including HR-UPLC/MS/MS, GNPS networking, SPME and GC/MS was employed to profile primary and secondary metabolites in both species that could mediate for new health and nutritive aspects, in addition to their aroma profiling that affects the consumer’s preferences.

According with my revision, the introduction should be improved substantially (in the last three paragraphs) (see the comments below).  In the Material and Method section (see the comments below). Results and discussion incorporate most of the ideas resumed in each topic.

The manuscript should be accepted with major revision.

Minor comments and questions

Abstract

Line 14: Improve the sentence of the main objective of the study.

Introduction

The introduction should be reformulated, because various paragraphs are part of the result section, please see the comments below:

Lines 50 to 62 and 68 to 74. Eliminate or modify these sentences because is part of the results section.

In Methods section:

This section provides most of the information about the data analyses, however, there is a lack of details that should be included, also I have some comments that should be addressed:

1)      Line 83: scientific names in cursive. In plant material indicate the origin and the number of the accession used for analyses. Please justify the decision to select these accessions.

2)      Also, answer this question. ¿How you define the samples for extract preparation?

3)      Section 2.2 Explain or give more details about the sample preparation.

4)      Line 97, 106, 112 and 123 mention the author of the references. Please review the format of citation.

5)      Line 107, include the reference literature used for.

6)      Why conduct the multivariate analysis (PCA, HCA and OPLS-DA). Please, incorporate this information in M&M section to clarify the methodology.

Results and Discussion section:

There is an error in the numbering of the pages. Please review it.

In section 3.1 there is a mistake? Metabolome profiling of C. pepo?. Please modify.

Line 158 to 161 correspond to methodology.

Please move the Figure 2, after Figure 1.

Figure 2, please review the tittle because you mention C. sativa and pumpkin.

“for L. siceraria (bottle gourd) and C. sativa (cucumber)”

 “The network is displayed as pie chart with orange and green colors representing distribution of the precursor ion intensity in the pumpkin”

Line 118, specify L. siceraria and C. pepo

Line 181-182, the author mentioned in accordance with references and databases, however, there are not references.

For the sentences in lines 238 to 245, please incorporate some references.

Line 302 – 303, its important to mention what type of bottle gourd, L. siceraria var. siceraria or L. siceraria var. hispida?

Line 340, please incorporate a reference for this sentence.

Conclusion section:

Very extensive. Please resume it.

Author Response

please see attached file addressing all comments

Reviewer 2 Report

How does Lagenaria siceraria (bottle gourd) metabolome compare to Cucumis sativus (cucumber) F. Cucurbitaceae? A multiplex approach of HR-UPLC/MS/MS and GC/MS using molecular networking and chemometrics

 Comments

The paper investigates the phytochemical composition of the cucumber (Cucumis sativus) and bottle gourd (Lagenaria siceraria) fruits, using HPLC-MS and GC-MS and implements principal component analysis for metabolome heterogeneity assessment. Additionally, the research team presents thorough aroma profiles of both fruits utilizing the Headspace solid phase microextraction technique followed by GC-MS analysis. The study showed the nutritional variations among the studied fruits. In addition, the investigation provides fundamental information on the metabolic profiles of both fruits for quality control purposes.

The work corresponds to the goal of Foods and should gain its readers appeal and attention. The manuscript is well edited with a good command of English language presenting the ideas clearly without ambiguity or grammatic mistakes.

Title:

The title of the manuscript is concise, descriptive, and emphasizes the key findings.

Abstract:

The abstract introduces a summary of the work content and represents a guide to the most important parts.

Pg 1, Ln 18: please add comma after ʻSPMEʼ

Introduction:

The Introduction section sets the context for the research work and highlights how it contributes to the knowledge in food chemistry field and builds on previous similar studies.

Pg 2, Ln 59: please remove the extra space between ʻusingʼ and ʻmetabolomicsʼ

Materials and methods:

Materials and methods section describes the material and design of the study in detailed way to make the work reproducible.

Results and discussion:

The section is presented in a logical sequence and summarizes the results, showing the major identified metabolites classes and explains the evidence of their structure elucidation.

Pg 4, Ln 163: please make the word ʻFigure 1ʼ non-bold.

Pg 4, Ln 171: please make the word ʻ Figure 2 and Figure S1ʼ non-bold.

Pg5, Ln 192: please change the word ʻptercarpansʼ to ʻpterocarpansʼ

Pg 5, Ln 195: please make the word ʻFigure 2ʼ non-bold.

Pg 5, Ln 197: please make the word ʻ Figure S1ʼ non-bold.

Pg 16, Ln 81: please remove the space in ʻhydroxymethylphenyl -O-hexosylferuloyl-O-malonylhexosideʼ to be ʻhydroxymethylphenyl-O-hexosylferuloyl-O-malonylhexosideʼ.

Pg 16, Ln 105: please make the word ʻTable 1ʼ non-bold.

Pg17, Ln 156: please change the word ʻptercarpansʼ to ʻpterocarpansʼ

Pg18, Ln 169: please remove the extra space between ʻidentified asʼ and ʻhedysarimpterocarpeneʼ

Pg10, Ln 369: please change the word ʻptercarpansʼ to ʻpterocarpansʼ

Conclusion:

The conclusion focuses on the main findings of the study, i.e. the metabolome/nutritive variations among the studied legume sprouts.

Tables and figures:

Tables and figures supplement the manuscript text, containing a huge set of data with simple clear presentation.

Pg 4-5, Ln 174-177: Figure 1 and its legend should be arranged alphabetically as the remaining order in the manuscript, where Cucumis sativa should precede Lagenaria siceraria.

Acknowledgments:

Pg 13, Ln 446-448: please add acknowledgment if necessary or remove the section.

References:

References are cited in a good way and match the style stated by the journal.

Pg 13, Ln 480: please make the plant name ʻPrunus armeniacaʼ italic to be ʻPrunus armeniacaʼ.

Pg 15, Ln 563: please make the plant name ʻCucumis meloʼ italic to be ʻCucumis meloʼ.

Therefore, I recommend it for publication with minor changes.

Author Response

(The authors gave the same response as above.)

Reviewer 3 Report

The manuscript is well documented. Just a few corrections

1) Line 107: The mass accuracy should less or equal to 5 ppm.

2) Line 156: The title of the paragraph is wrong. It referes to C. pepo and C. sativus instead of Lagenaria siceraria and Cucumis sativus.

Author Response

(The authors gave the same response as above.)
